# Functional Annotation of Bacterial Signal Transduction Systems: Progress and Challenges

**DOI:** 10.3390/ijms19123755

**Published:** 2018-11-26

**Authors:** David Martín-Mora, Matilde Fernández, Félix Velando, Álvaro Ortega, José A. Gavira, Miguel A. Matilla, Tino Krell

**Affiliations:** 1Department of Environmental Protection, Estación Experimental del Zaidín, Consejo Superior de Investigaciones Científicas, Prof. Albareda 1, 18008 Granada, Spain; david.martin@eez.csic.es (D.M.-M.); mtdfernandez@gmail.com (M.F.); felix.velando@eez.csic.es (F.V.); 2Department of Biochemistry and Molecular Biology ‘B’ and Immunology, Faculty of Chemistry, University of Murcia, Campus of Espinardo, Regional Campus of International Excellence “Campus Mare Nostrum”, 30100 Murcia, Spain; alvarort@um.es; 3Laboratorio de Estudios Cristalográficos, IACT, (CSIC-UGR), Avenida las Palmeras 4, 18100 Armilla, Spain; jgavira@iact.ugr-csic.es

**Keywords:** bacterial signal transduction systems, chemotaxis, transcriptional regulators, chemoreceptors, sensor kinases

## Abstract

Bacteria possess a large number of signal transduction systems that sense and respond to different environmental cues. Most frequently these are transcriptional regulators, two-component systems and chemosensory pathways. A major bottleneck in the field of signal transduction is the lack of information on signal molecules that modulate the activity of the large majority of these systems. We review here the progress made in the functional annotation of sensor proteins using high-throughput ligand screening approaches of purified sensor proteins or individual ligand binding domains. In these assays, the alteration in protein thermal stability following ligand binding is monitored using Differential Scanning Fluorimetry. We illustrate on several examples how the identification of the sensor protein ligand has facilitated the elucidation of the molecular mechanism of the regulatory process. We will also discuss the use of virtual ligand screening approaches to identify sensor protein ligands. Both approaches have been successfully applied to functionally annotate a significant number of bacterial sensor proteins but can also be used to study proteins from other kingdoms. The major challenge consists in the study of sensor proteins that do not recognize signal molecules directly, but that are activated by signal molecule-loaded binding proteins.

## 1. Introduction

Bacteria have evolved an array of different signal transduction mechanisms that are able to sense and respond to a wide range of environmental cues and signal molecules. These systems are generally transcriptional regulators, two-component systems (TCS) and chemosensory pathways [1,2,3] (Figure 1).

Typically, the capacity of transcriptional regulators to regulate promoter activity is modulated by the recognition of signal molecules. Alternatively, TCS function is based on the signal mediated modulation of the sensor kinase activity that in turn modulates transphosphorylation kinetics to the response regulator. As in the case of transcriptional regulators, the majority of bacterial TCSs appear to be involved in transcriptional regulation [4,5]. Chemosensory pathways can be understood as sophisticated versions of TCSs where the stimulus is received by chemoreceptors that in turn modulate the activity of the CheA histidine kinase leading to alterations in CheY phosphorylation. The majority of chemosensory pathways mediate chemotaxis, whereas others carry out alternative cellular functions or are associated with type IV pili based motility [2,6].

There is an enormous diversity in the domain organization and topology of signal transduction systems [1,3,6,7,8]. This diversity is also reflected in a variety of different mechanisms that regulate receptor activity, such as stimulus mediated alterations in the transmembrane regions of sensor kinases and chemoreceptors [9,10], signal sensing by the cytoplasmic autokinase domain of histidine kinases [11], chemoreceptor activation by proteolysis [12] or via the phosphotransferase system [13].

However, the canonical mode of their activation consists in the binding of signal molecules to the ligand binding domains (LBD) that are present in all three major signal transduction systems (Figure 1). Thus, signal binding to many transcriptional regulators modulates their affinity for promoter regions [14]. In contrast, ligand binding to the LBD of sensor kinases [15] and chemoreceptors [16] was shown to cause piston-like movements of transmembrane regions that ultimately cause changes in autokinase activity. This appears to be a general mechanism that applies to sensor kinases and chemoreceptors, since, firstly, chimeric receptors comprising different types of chemoreceptor LBDs with the cytosolic fragment of the Tar chemoreceptor were functional [17,18] and, secondly, a number of functional chemoreceptor/sensor kinase chimeras were produced [19,20].

The majority of bacterial sensor proteins are of unknown function. For other systems, a function has been identified mainly through the phenotypic characterization of mutant strains. However, in a significant number of cases information on the stimulus that modulates the activity of a given system is lacking. This lack of knowledge on signals recognized by sensor proteins represents a major bottleneck in signal transduction research [21]. This can be best illustrated by the three sensor kinases GacS, LadS and RetS. Although these proteins play a central role in the virulence of the human pathogen *Pseudomonas aeruginosa* [22], the signals that stimulate these three proteins have so far not been identified.

Sensor protein function is closely related to the signals recognized by the corresponding LBD. However, LBDs show important sequence diversity and, as a result, the ligand specificity of a given LBD is frequently not reflected in overall LBD sequence homology [23], which in turn hampers functional annotation by extrapolation from homologous systems. This limitation makes experimental approaches for the functional annotation of bacterial sensor proteins essential. In this article we will discuss progress made over mainly the last 10 years in experimental approaches that resulted in the functional annotation of a significant number of bacterial sensor proteins.

## 2. Functional Annotation Using Genetic Approaches

Insight into the function of genes and proteins can be gained by the phenotypic analysis of bacterial chemoreceptor mutants. Using this approach the function of many chemoreceptors has been identified. As representative examples we would like to cite here the identification of chemoreceptors for naphthalene [24], cyclic carboxylic acids [25], cytosine [26], inorganic phosphate [27] or boric acid [28]. However, this strategy has several limitations that we illustrate here.

### 2.1. Multiple Receptors

Chemotactic bacteria possess on average 14 chemoreceptor genes [29] and in some case up to 80 genes were detected [30]. There are a number of reports showing that some species possess multiple receptors that respond to the same ligand. In these cases the loss of activity caused by the mutation of a single chemoreceptor gene may be compensated by additional receptors and the analysis of chemoreceptor single mutants may not lead to a functional annotation. For example, wild type *Comamonas testosteroni* and a mutant defective in the chemoreceptor Mcp2983 had indistinguishable chemotaxis to a series of organic acids and organic compounds. However, the complementation of a chemotaxis free mutant, in which all 22 chemoreceptor genes were deleted, with the *mcp2983* gene resulted in the recovery of wild-type like chemotaxis to a number of chemoeffectors [31]. The authors conclude that the genome of *C. testosteroni* encodes additional chemoreceptors that compensate the deletion of the *mcp2983* gene. Another example is *Ralstonia pseudosolanaceum* that was identified to contain at least two chemoreceptors for citrate. In analogy to the above study, single mutants in each of the two receptors did not alter citrate chemotaxis and a reduction was only observed in the double mutant [32]. Another recent study revealed that chemotaxis to histamine in *P. aeruginosa* is mediated by the combined action of three chemoreceptors, TlpQ, PctA and PctC. TlpQ binds histamine with very high affinity and a mutant in *tlpQ* was only defective in histamine chemotaxis at low concentrations of the chemoattractant [33]. Also, the recent analysis of the chemoreceptor repertoire of *Bacillus amyloliquefaciens* identified multiple receptors that responded to a same amino acid, sugar or organic acid [34]. Further examples are multiple amino acid receptors in *P. aeruginosa* [35,36], *P. fluorescens* [37] and *Sinorhizobium meliloti* [38,39] as well as several chemoreceptors in *P. fluorescens* Pf0-1 or *P. putida* KT2440 that respond to Krebs cycle intermediates [40,41,42,43].

### 2.2. Chemotaxis Is Induced or Repressed by the Cognate Ligands

There is also evidence that some chemoeffectors either induce [25] or repress [27,44] the chemotaxis phenotype. For example, *P. aeruginosa* does not show any chemotaxis towards inorganic phosphate (Pi) under standard culture conditions in which cells are grown in media containing significant amounts of Pi [27,44]. However, very strong Pi chemotaxis was observed under Pi limiting conditions, which is due to the fact that Pi represses the expression of its cognate chemoreceptors [27,45].

### 2.3. Energy Taxis May Mask Chemotaxis

Tactic movements can be due to chemotaxis, typically characterised by the recognition of the chemoeffector by a chemoreceptor in the extracytosolic space, or energy taxis, which is based on sensing molecular consequences that occur as a result of chemoeffector metabolization [46]. However, in a number of cases chemo- and energy taxis to a given compound occur simultaneously [47,48]. For example, malate energy taxis in *P. aeruginosa* was found to dominate and mask malate chemotaxis to an extent that chemotaxis became only visible in a mutant defective in the energy taxis chemoreceptor [48]. Therefore, as concluded before, the phenotypic characterization of chemoreceptor single mutants may not result in identifying the function of these receptors.

## 3. Functional Annotation Using Thermal Shift Assays of Purified Recombinant Sensor Proteins

An alternative to the annotation of sensor proteins using genetic approaches is the in vitro screening for ligands that bind to purified sensor proteins or individual LBDs. This method is based on the fact that the binding of ligands to proteins either increases or decreases their thermal stability. Typically Differential Scanning Fluorimetry [49] is used for Thermal Shift Assays. In these assays, a hydrophobic fluorescent dye is added to the protein that is then exposed to a temperature gradient. In soluble proteins, hydrophobic amino acids are typically in the protein interior and excluded from the solvent. However, during temperature-induced protein unfolding, these amino acids are exposed to the solvent leading to additional fluorescent dye binding altering in turn protein fluorescence. From the recorded fluorescence changes, the melting temperature (Tm value) can be derived, which corresponds to the temperature at which half the protein is unfolded, whereas the remaining half is still in its folded state. Typically, the binding of a ligand to a protein retards protein unfolding and as such increases the Tm value. However, in some cases a reduction in Tm is observed indicating that the ligand binds to an unfolded state of the protein [50,51]. There are cases where the binding of different ligands to the same protein causes either increases or decreases of the thermal stability [50]. A major advantage of this assay is that it can be used in high-throughput format using 96-well plates and commercially available compound collections. In general, Tm increases superior to 2 °C are considered significant. This method was first reported as an high-throughput drug screening approach [52], but was adapted by McKellar et al. [53] for the identification of ligands that bind to LBDs.

Most sensor kinases and chemoreceptors are transmembrane proteins, which are rather difficult to work with at the biochemical level. However, there is now solid evidence showing that most of their LBDs can be generated as individual, recombinant proteins that maintain their binding capacity for ligands. Therefore, this approach can be either used for full-length transcriptional regulators or individual LBDs. In recent years this approach has been used to annotate a significant number of bacterial sensor proteins (Table 1). Further, more technical information on this method can be found in [54,55].

### 3.1. The Case of The Transcriptional Regulator AdmX

We would like to illustrate this approach on the example of the transcriptional regulator AdmX of *Serratia plymuthica* A153 [56]. This strain is a root-associated bacterium that serves as a model strain for the investigation of the biosynthesis and regulation of secondary metabolites [57]. Among the secondary metabolites synthesized by this bacterium is andrimid, an antibiotic that inhibits the bacterial acetyl-CoA carboxylase [58]. Proteins required for andrimid biosynthesis are encoded in the adm gene cluster of around 25 kbp that is predicted to contain at least 21 ORFs [58]. Upstream of this cluster is a gene encoding a 304 amino acid transcriptional regulator, termed AdmX. Interestingly, the deletion of *admX* resulted in a complete abolition of andrimid production [58]. What is thus the molecular mechanism by which AdmX controls andrimid production? First clues can be derived from a sequence analysis showing that AdmX is predicted to contain two domains, a helix-turn-helix (HTH) motif containing DNA binding domain and a LBD (Figure 2A).

AdmX was thus hypothesized to modulate gene expression in response to ligands that interact with its LBD. The central question in understanding AdmX-mediated function was to identify the ligands that are recognized by its LBD.

#### 3.1.1. Thermal Shift Assays with AdmX and AdmX-LBD

To address this question, full-length AdmX as well as its individual LBD, AdmX-LBD (Figure 2A), were produced as purified recombinant proteins and submitted to Thermal Shift Assays. Since AdmX-LBD showed higher solubility and yields than the full-length protein, Thermal Shift Assays were initiated using the individual LBD. The unfolding curve of ligand-free AdmX-LBD is shown in Figure 2B and the minimum of the first derivatives of these data (Figure 2D) indicates that the Tm of the protein is 56.5 °C. The screening of approximately 1700 compounds showed that the unfolding behaviour of AdmX-LBD in the presence of the auxin indole-3-acetic acid (IAA) was different, since the Tm was increased by 5.1 °C (Figure 2B,D). We subsequently investigated the effect of IAA on the thermal unfolding of the full-length protein (Figure 2C,E). Two unfolding events could be distinguished centred at Tm values of 44.6 and 57.5 °C, which correspond to the sequential unfolding of the HTH and LBD domains, respectively. In the presence of IAA, the Tm of the LBD domain was increased in a very similar manner (shift of 6 °C) (Figure 2C,E). Interestingly, the unfolding of the HTH domain was also slightly increased (shift of 1.4 °C), indicative of IAA-mediated domain cross-talk (Figure 2C,E).

#### 3.1.2. Study of Ligand Binding to AdmX and AdmX-LBD by Isothermal Titration Calorimetry

To characterise the interaction of IAA with AdmX and AdmX-LBD, we carried out Isothermal Titration Calorimetry (ITC) experiments (Figure 2F,G). Initial experiments involved the titration of buffer with IAA to quantify dilution heats (note: downwards going peaks indicate exothermic reactions, whereas upwards going peaks represent endothermic binding). IAA bound to AdmX-LBD with a *K*_D_ of 15 µM, whereas binding to the full-length protein showed a lower affinity (*K*_D_ = 61 µM). Interestingly, IAA binding to AdmX-LBD was exothermic (favourable enthalpy changes) whereas binding to full-length AdmX was endothermic indicative of unfavourable enthalpy changes. Both binding events were thus driven by favourable entropy changes, which is likely due to the displacement of a significant amount of protein-bound water to the disordered bulk water pool. The thermodynamic differences in binding at AdmX and AdmX-LBD are likely due to enthalpy changes that arise from the above mentioned intra-domain communication in the full-length protein.

#### 3.1.3. The Identification of the AdmX Signal Molecule Permits to Elucidate the Molecular Mechanism and Physiological Relevance

Subsequent in vivo studies using antibiosis assays showed that the addition of IAA to bacterial cultures reduced andrimid production in a dose-dependent manner [56]. AdmX was found to bind to the promoter upstream the first gene of the adm gene cluster and in vitro transcription experiments revealed that IAA reduced the AdmX mediated expression from this promoter. These experiments were consolidated by in vivo gene expression studies [56].

IAA is an universal signal molecule that is synthesized not only by plants and bacteria but also by archaea, fungi and animals [56,59]. Remarkably, IAA produced by other plant-associated bacteria was shown to repress andrimid production in *S. plymuthica* A153, as an indication that this IAA-mediated process causes inter-species communication to modulate antibiotic synthesis [56].

### 3.2. Functional Annotation of Chemoreceptors Using Thermal Shift Assays

Over the last years, Thermal Shift Assays have been used to functionally annotate a number of chemoreceptors and the corresponding data are summarized in Table 1. These studies were conducted using the individual periplasmic LBDs and, in all cases, the corresponding chemoreceptors were found to mediate chemoattraction. Importantly, these studies have permitted to identify chemoreceptors with novel ligand profiles such as the first chemoreceptor for quaternary amines [39], purines [51] or polyamines [60]. Protein stability in the absence of ligands ranged from 36 to 57 °C (average 46 ± 6 °C), permitting protein handling at room temperature. Tm shifts of verified binding events ranged from 1.1 °C to 14 °C, with an average shift of 6.7 ± 4 °C (Table 1).

Several conclusions can be derived from these data.

(1) The receptors PscD and McpV are homologues and belong together with McpP [61] to the sCACHE domain containing chemoreceptor family for small organic acids like acetate, propionate or pyruvate. PscD-LBD and McpV-LBD are similar in size, share 43% of sequence identity and possess almost identical thermal stabilities (Table 1). However, the effect of their ligands on protein thermal stability is very different. Whereas the binding of the above 3 ligands to PscD-LBS causes very modest shifts of 1 to 3.3 °C, the Tm shifts caused by the same ligands to McpV are in the double digit range (Table 1).

(2) In general, Tm shifts superior to 2 °C are considered significant for compound selection for further studies. Although valid for most of the cases, data shown in Table 1 illustrate that this is to be considered with caution. As for PscD-LBD, glycolate caused a Tm shift of merely 1.1 °C, but was the ligand that bound with highest affinity to the protein.

(3) Several studies reported compounds that shifted the Tm but that did not show binding in ITC. It was hence concluded that these compounds do not bind to the protein and were referred to as false positive hits. However, this has not necessarily to be the case. Due to the limitation of ligand dilution heats caused by the injection of elevated ligand concentrations [62], ITC is frequently not suited to monitor low-affinity binding events. The failure to observe ligand binding by ITC implies that there is no high affinity binding but does not exclude that ligands may bind with lower affinities that are not detectable by ITC.

## 4. Three-Dimensional Structural Information Provides Clues on Ligands Recognized by Homologous Sensor Proteins

Ligand binding domains are characterized by a high degree of sequence divergence and frequently the overall sequence similarity between homologous proteins is not reflected in a similarity of ligands recognized [23]. However, three-dimensional structural information is now an invaluable tool to get initial information on ligands recognized. We would like to illustrate this issue on the example of CACHE domains that exist either in a mono-modular (sCACHE) or bi-modular (dCACHE) configuration [68]. CACHE domains are the most abundant sensor domains in both, histidine kinases [69] and chemoreceptors [7].

As significant number of structures deposited in the Protein Data Bank (pdb) are the result of structural genomics initiatives and represent a source of information on the function of homologous proteins. In the framework of structural genomic projects on *Anaeromyxobacter dehalogenans* and *Vibrio parahaemolyticus*, structures of sCACHE domains have been deposited at pdb (pdb ID 4K08 and 4EXO, respectively). In both cases, their binding pocket was occupied by a ligand, namely acetate in 4K08 and pyruvate in 4EXO. In the case of the 4K08 structure, acetate was a component of the crystallization buffer. Since pyruvate was absent from the crystallization buffer of the 4EXO structure, the ligand must have been co-purified with the protein. *P. putida* KT2440 has a single chemoreceptor with a sCACHE domain (PP_2861), which shares only 22% sequence identity with the 4EXO protein. The above observations have directed our research and we have tested whether the LBD of PP_2861 also bound acetate and pyruvate. We were able to show that this was the case and have identified with propionate and L-lactate two additional ligands. The corresponding chemoreceptor was renamed McpP and was found to mediate chemoattraction to these four ligands [61].

Similarly, structural information was also useful to study dCACHE domain. The first structure of a dCACHE domain deposited into pdb was entry with ID 3C8C; solved in a structural genomics project of *Vibrio cholerae*. In analogy to the case above, the binding pocket of this structure was occupied with a ligand, l-Ala, and since l-Ala was not present in the crystallization buffer, it must have been co-purified with the protein suggesting that it is a physiological ligand. This information has been useful to identify other amino acid sensing chemoreceptors and it was established that dCACHE containing chemoreceptors form the primary family of amino acid responsive chemoreceptors [18].

There are different sub-families within dCACHE domains, namely those that bind C4-dicarboxylic acids like succinate and malate [70,71] or those that bind different amines such as amino acids, GABA, polyamines, taurine, purines or quaternary [23,33,36,39,51,60,72,73,74,75]. Structural information has now provided the molecular detail of C4-dicarboxylic acid [70,71] and amine recognition at different dCACHE domains [33,73,74,75] (Figure 3). In all cases the ligand is bound to the membrane distal module. The comparison of amine recognition patterns in structures from different species (*P. aeruginosa*, *P. putida*, *V. cholerae*, *Campylobacter jejuni*), and in complex with different ligands (putrescine, histamine, taurine, amino acids) identifies the consensus motif Y-x-d-x(n)-d that coordinates the amino group of ligands through a hydrogen bonding network. Therefore, the sequence analysis of dCACHE domains of unknown function for the presence of this motif can provide first clues on the ligand recognized.

## 5. Protein Structure Based Virtual Ligand Screening

Initial clues on ligands recognized by bacterial sensor proteins can also be obtained by virtual ligand screening. This approach requires a structural model of the corresponding protein or domain as well as knowledge on the location of its ligand binding site. ZINC is a free database of three-dimensional structures of commercially available compounds that can be used for virtual screening [77,78]. In silico docking experiments can be conducted in high-throughput format using subsets of the ZINC database to the target proteins using programs like GLIDE [79]. The quality of the ligand-protein fit will be expressed by a docking score that corresponds to protein-ligand free energy estimations, which were found to be similar to experimentally determined values [80]. Compounds that result in low docking scores are thus candidates for further studies.

### 5.1. The Identification of Rosmarinic Acid as Plant Derived Quorum Sensing Agonist

The usefulness of this approach is here illustrated on the example of bacterial quorum sensing (QS), which is based on the synthesis and detection of QS molecules. This strategy allows bacteria to coordinate gene expression in a cell-density manner and consequently to regulate processes that are beneficial when performed by groups of bacteria acting in synchrony [81]. QS is a mechanism for inter- and intra-species bacterial communication, but there are an increasing number of reports showing that plants produce compounds that interfere with bacterial QS by binding to bacterial QS receptors [82]. Nonetheless, very little is known about the identity of these plant-derived compounds. *P. aeruginosa* produces homoserine lactone QS signal molecules through two homoserine lactone synthases, RhlI and LasI. These signal molecules are subsequently sensed by three transcriptional regulators that belong to LuxR family, RhlR, LasR and QscR that are composed of a LBD and a DNA binding domain.

#### 5.1.1. Identification of Rosmarinic Acid as RhlR Ligand

To identify potential plant derived compounds that may bind to two of these receptors, RhlR and LasR, virtual docking studies to a structural model of the RhlR-LBD and the structure of LasR-LBD (PDB ID: 3IX3) were conducted using the Natural Compounds Subset of the ZINC database (containing 5391 compounds at the moment of analysis) [83,84]. Nine plant-derived compounds with a low docking score were selected for microcalorimetric titrations of purified RhlR and LasR [83,85]. Initial experiments demonstrated that both proteins were able to recognize different homoserine lactones. However, none of the plant-derived compounds bound to LasR, but rosmarinic acid (RA) was found to bind to RhlR with nanomolar affinity [83]. In vitro transcription assays demonstrated that RA enhances transcription from an RhlR controlled promoter; a finding that was then consolidated by in vivo gene expression experiments. The addition of RA to the growth medium induced phenotypes that are typically controlled by QS, such as the production of the virulence factor pyocyanin, biofilm formation or elastase activity [83]. The positions of bound RA and 3-oxo-C12-homoserine lactone in the superimposed LBDs of RhlR and LasR are shown in Figure 4.

#### 5.1.2. Assessment of the Global Role of Rosmarinic Acid

RNA-seq experiments were conducted to assess the global effect of RA on *P. aeruginosa* [86]. In the presence of RA the transcript levels of 128 genes were upregulated in an *rhlI/lasI* mutant that is unable to produce homoserine lactones. Interestingly, 88% of these genes have been reported previously to be regulated by QS. No RA-mediated increase in gene expression was observed in the *rhlI/lasI*/*rhlR* triple mutant, confirming that the phenotypic consequences of RA are due to RhlR binding. We concluded that the plant compound RA induces a broad QS response in *P. aeruginosa*, which corresponds to a novel mechanism in inter-kingdom signaling [86].

In summary, the output of these screening experiments are relatively noisy, since microcalorimetric titration of most of the selected compounds did not reveal high-affinity binding. However, within these nine compounds was a single compound, RA that did bind. Using alternative experimental strategies it would have been very difficult to detect this QS agonist since previous studies have shown that RA is not always present in root exudates, which are typically used for the study of plant derived compounds that interfere with QS [87]. In sweet basil the infection of plants by *P. aeruginosa* was shown to trigger the root secretion of RA, which was undetectable prior to infection [87].

### 5.2. Further Examples that Illustrate the Usefulness of Virtual Ligand Screening to Identify Protein Ligands

A number of in silico docking experiments have been reported that have led to the identification of ligands for a number of bacterial signal transduction systems. Two studies report the identification of ligands that bind to the transcriptional regulators AphB of *V. cholerae* [88,89] that regulates the expression of genes encoding cholera toxin and toxin-co-regulated pilus. In both cases virtual docking results were verified experimentally by NMR [88] and/or ITC [89]. One of the compounds identified, ribavirin, was shown to inhibit cholera toxin production, which in turn was reflected in a reduction of intestinal colonization and intracellular survival of *V. cholerae* [89].

The sensor kinase KinB controls the synthesis of the exopolysaccharide alginate in *P. aeruginosa*. It has a periplasmic LBD that forms a four-helix bundle structure, but the nature of ligands that bind to KinB-LBD is unknown. In silico ligand screening assays were conducted that resulted in the identification of several sugar phosphates, including compounds that serve as precursors and intermediates of the alginate biosynthesis. The binding of two of these compounds was verified experimentally using biolayer interferometry [90].

AldR is a transcriptional regulator of *Mycobacterium tuberculosis* and controls the expression of the *ald* genes encoding the alanine dehydrogenase that plays a major role in the response to nutrient starvation. AldR is composed of a DNA binding domain and an AsnC type LBD [91]. To identify AldR ligands, in silico screening assays identified a tetrahydroxyquinolone carbonitrile derivative. This compound was found to bind to purified AldR and to prevent protein binding to promoter DNA [91].

LdtR is a transcriptional regulator from *Liberibacter asiaticus*, a non-culturable citrus fruit pathogen. Previous work using thermal shift assays has resulted in the identification of several Ltd.R ligands that were found to reduce LdtR-DNA binding [92]. To study of the effect of the compounds identified, the authors used *S. meliloti* and *Liberibacter crescens* that possess LdtR homologues and showed that molecules identified decrease their stress tolerance. However, no information was available on the binding site of these ligands at LdtR. In silico docking studies with one of the molecules identified, benzbromarone, identified its binding pocket at LdtR for [93]; a finding that was verified by site-direct mutagenesis. The gained knowledge will be useful for the design of therapeutics to fight this pathogen.

## 6. The Challenge: Signal Input Through Ligand Binding Proteins

### 6.1. Sensor Protein Activation by the Binding of Ligand Binding Proteins

In our laboratory we have isolated a large amount of individual chemoreceptor LBDs, mostly from different *Pseudomonas* species. Although these proteins were folded in most of the cases, screening for directly binding ligands did not provide any hit for a considerable number of cases. This could be due to the possibility that the ligands of a given LBD were not amongst the ligands screened or that the protein was incorrectly folded. However, what might be more likely is the possibility that sensor kinases and chemoreceptors can also be activated by the binding of ligand-loaded binding proteins to the LBD.

In fact, there is increasing evidence for ligand binding protein mediated activation of chemoreceptors and sensor kinases. Thus, three of the four *Escherichia coli* chemoreceptors are activated by ligand binding proteins that contain sugars or dipeptides [94,95,96]; the Pi responsive CtpL chemoreceptor of *P. aeruginosa* is activated by the PstS ligand binding protein [44]; chemotaxis to the autoinducer-2 required the Tsr receptor and the periplasmic binding protein LsrB in *E. coli* [97] and the TlpB receptor as well as the AibA and AibB ligand binding proteins in *Helicobater pylori* [98]. Further studies indicate that binding proteins may stimulate chemoreceptors in *Bacillus subtilis* [99] and *C. jejuni* [100]. Based on sequence classification, chemoreceptors were found to employ more than 80 different LBD types [7] and indirect ligand binding mechanism do not appear to be restricted to a given LBD type since the *E. coli* receptors possess a 4-helix bundle (4HB) LBD [101], CtpL a helical bimodular (HBM) domain [102], TlpB a sCACHE domain [68] and the *B. subtilis* and *C. jejuni* receptors possess dCACHE domains [68].

Binding proteins also stimulate sensor kinases and high resolution structural information is available for the interaction of d-xylose-loaded XylFII with LytS-LBD [103], trimethylamine-bound TorT with TorS-LBD [104] and the autoinducer-2/LuxP complex with LuxQ-LBD [105] (Figure 5). The relatively low number of indirect binding mechanisms so far identified may rather be due to the technical complexity to identify such mechanisms than to a low occurrence in nature.

### 6.2. Identification of Ligand Binding Proteins That Interact With Sensor Proteins

Approaches for large scale characterization of protein-protein interactions include yeast two-hybrid screening [106], co-immunoprecipitation [107], affinity purification-mass spectrometry [108], in vitro cross linking-mass spectrometry [109], phage display [110], yeast display [111] or reprogramming yeast mating [112].

The identification of the ligand binding proteins is here illustrated on the CtpL chemoreceptor of *P. aeruginosa.* The deletion of the *ctpL* gene abolished chemotaxis to low Pi concentrations [27], but microcalorimetric titration of CtpL-LBD with Pi did not provide any evidence for direct binding. It was hence hypothesized that CtpL may be stimulated by a periplasmic ligand binding protein. Immobilized CtpL-LBD was brought in contact with an extract of soluble *P. aeruginosa* proteins. Following a washing step, bound proteins were eluted and the single major band on the SDS-PAGE gel of the eluate was identified by mass spectrometry as PstS [44]. This protein interacts with the Pi uptake system PstABC, providing the ligand to be transported, and is also involved in transcriptional responses to changing Pi concentrations [113]. ITC studies revealed that PstS bound to CtpL-LBD and a mutant in *pstS* was deficient in chemotaxis to low Pi concentrations indicating that PstS is the only protein that stimulates CtpL [44]. PstS has thus a triple function which consists in Pi uptake, transcriptional regulation and chemotaxis. For a significant number of chemoreceptor LBDs we have conducted similar pull-down experiments that have, however, not resulted in the identification of any protein. What may be possible reasons?

(1) Ligand binding proteins comprise a large protein family that share the same overall bilobal structure, but differ significantly in sequence, size and also in their mechanism since some bind to sensor proteins in both, the apo- and holo- form, whereas others only in their ligand bound state. Representative examples for the former group are the interactions of the LuxP [114], XylFII [103] or TorT [104] ligand binding proteins (Figure 5) with their respective sensor proteins that occur in both, the ligand-bound and ligand-free state. In contrast, only the maltose-bound form of the maltose-binding protein interacts with the LBD of the Tar chemoreceptor [115]. Therefore, a potential reason for the failure of pull-down experiments is that the cognate protein ligand was not present, hence preventing protein binding.

(2) Another possible reason for the failure of pull-down experiments is related to the observation that the expression of periplasmic binding proteins is frequently subject to a strict transcriptional control. There are cases where the cognate ligand either induced [116,117] or repressed [45,118] binding protein gene expression. For example, in *P. aeruginosa* the reduction of the Pi concentration in the growth medium from 1 to 0.2 mM increased transcript levels of the *pstABC* transporter genes 19 to 28-fold, whereas the transcript of the PstS Pi binding protein was increased 223-fold [45]. Based on this information, the above mentioned pull-down experiments were conducted with bacteria grown under Pi starvation conditions. When the same experiment was carried out in rich medium, containing significant amounts of Pi, no binding protein could be identified, since not present in the cell extract [44].

Taken together, pull-down experiments should possibly be conducted using cell extracts of bacteria grown in different growth media. If there is preliminary information available on potential ligands, these should be added to the cell extract and to the solutions used for the pull-down experiments.

## 7. Concluding Remarks

The individual LBDs of many bacterial sensor proteins can be obtained as soluble proteins that maintain their ligand binding properties. The analysis of such proteins using thermal shift based ligand screening can result in the identification of ligands, which permits the identification of protein function. Although this approach has primarily been applied to bacterial chemoreceptor research, it is also suitable to identify ligands for any other sensor protein. A major challenge resides in the study of signal transduction systems that are stimulated by ligand binding proteins. There is currently a lack of information on ligands recognized by different ligand binding proteins and a major research need consists in the functional annotation of these proteins. Such information will be useful to study the function of signal transduction systems that are stimulated by indirect binding mechanisms.

## Figures and Tables

**Figure 1 ijms-19-03755-f001:**
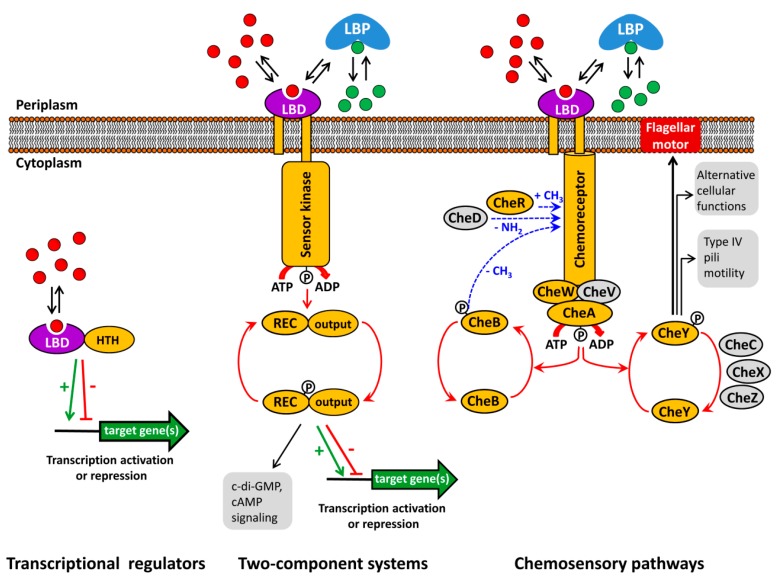
Major bacterial signal transduction systems. LBD: ligand binding domain; HTH: helix-turn-helix motif containing DNA binding domain; LBP: ligand binding protein; REC: receiver domain. The core proteins of chemosensory pathways, present in most pathways, are shaded in yellow, whereas auxiliary proteins, present only in some pathways, are shown in grey. red and green dots: signal molecules; red arrows represent phosphorylation/dephosphorylation events, blue arrows represent methylation/demethylation/deamidation of amino acids; black arrows indicate the pathway output, the green arrow and red T-bar indicate transcriptional activation or inhibition, respectively.

**Figure 2 ijms-19-03755-f002:**
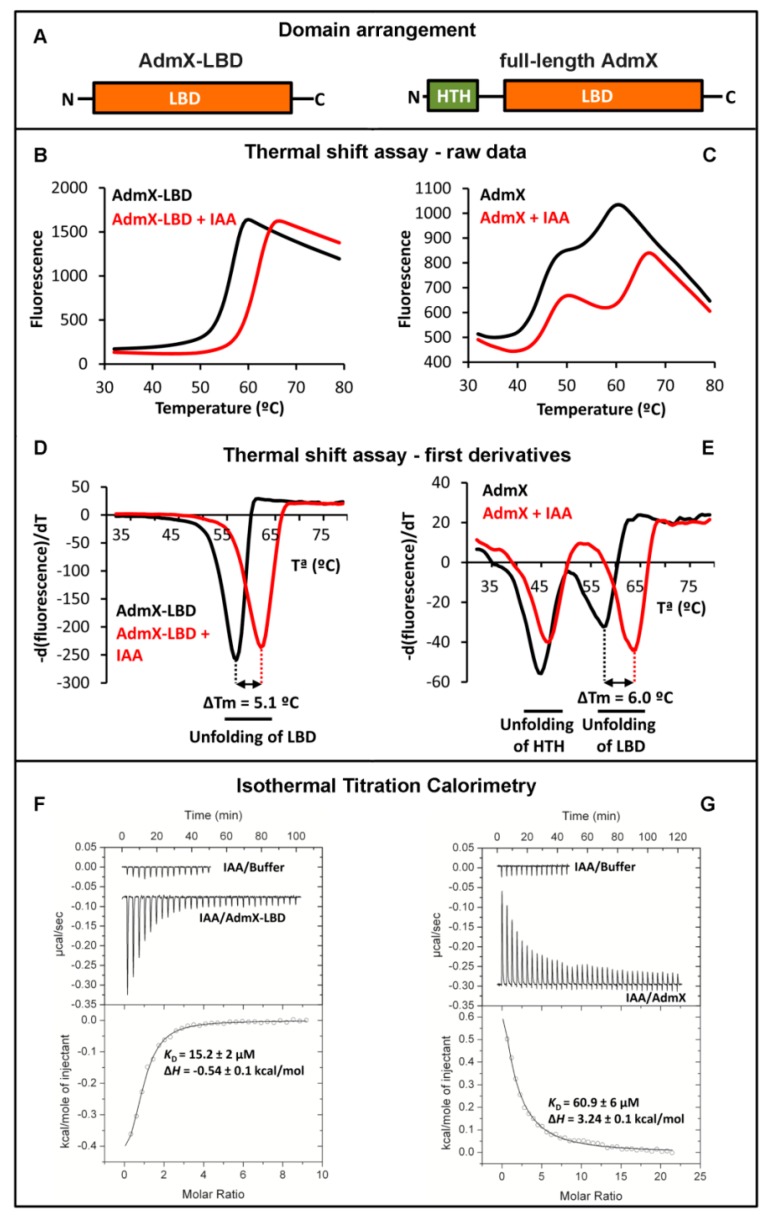
Use of the Thermal Shift Assay and Isothermal Titration Calorimetry to identify ligands that bind to AdmX. (**A**) Domain arrangement of AdmX and its ligand binding domain (LBD). (**B**–**E**) Thermal shift experiments of AdmX and AdmX-LBD in the absence and presence of indole-3-acetic acid (IAA). (**B**,**C**) raw data; (**D**,**E**) first derivatives of raw data; (**F**,**G**) microcalorimetric titrations of buffer, AdmX and AdmX-LBD with IAA. Upper panel: titration raw data; lower panel: fit of dilution heat-corrected and concentration-normalized raw data with a model for the binding of a single ligand to a macromolecule. The derived thermodynamic binding parameters are indicated.

**Figure 3 ijms-19-03755-f003:**
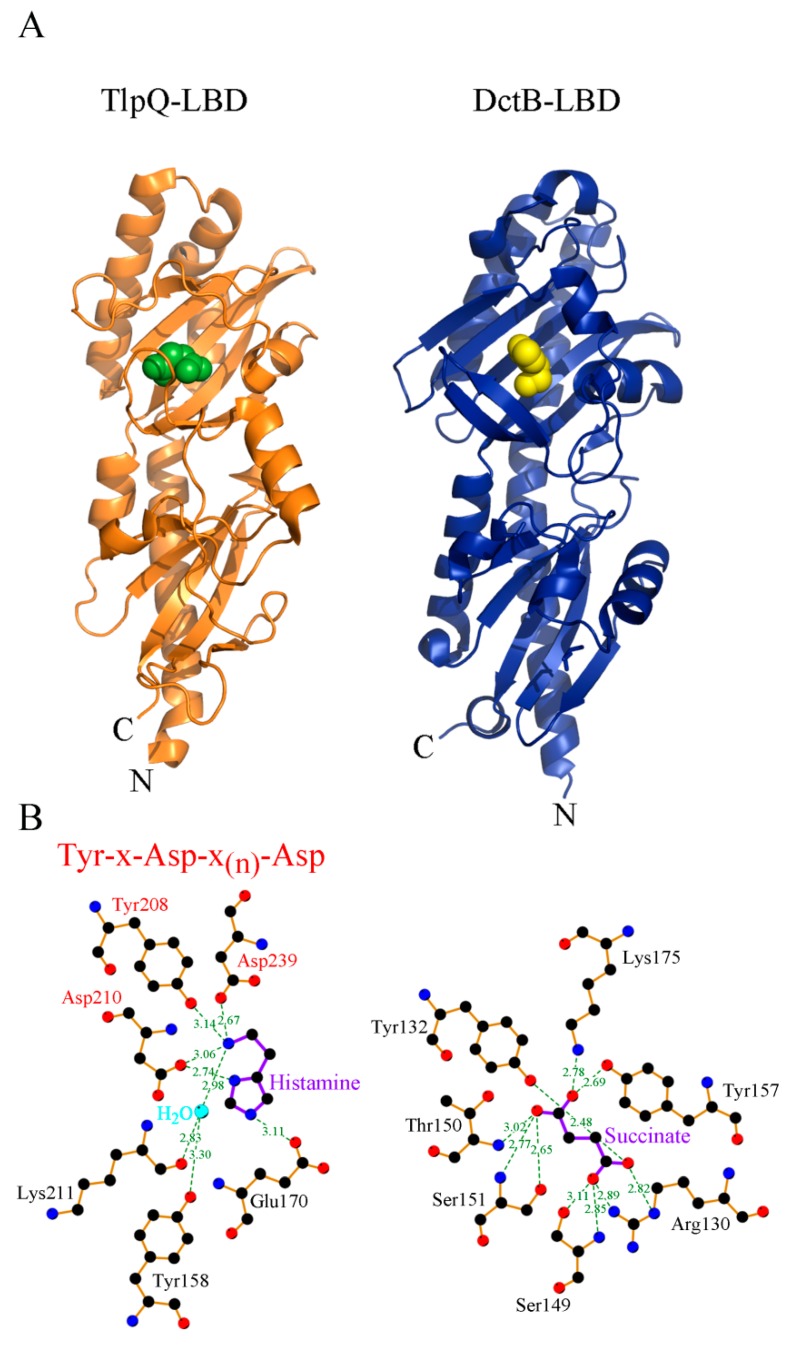
Three-dimensional structure and mode of ligand recognition at dCACHE domains. (**A**) Ribbon diagram of the LBD of the TlpQ chemoreceptor of *P. aeruginosa* (orange) in complex with histamine (green, pdb ID 6fu4) and LBD of the DctB sensor kinase of *V. cholerae* (blue) in complex with succinate (yellow, pdb ID 3by9). (**B**) Amino acids involved in ligand recognition. The sequence motif involved in amine recognition is shown in red above the histamine plot. Figure generated using Ligplot [76].

**Figure 4 ijms-19-03755-f004:**
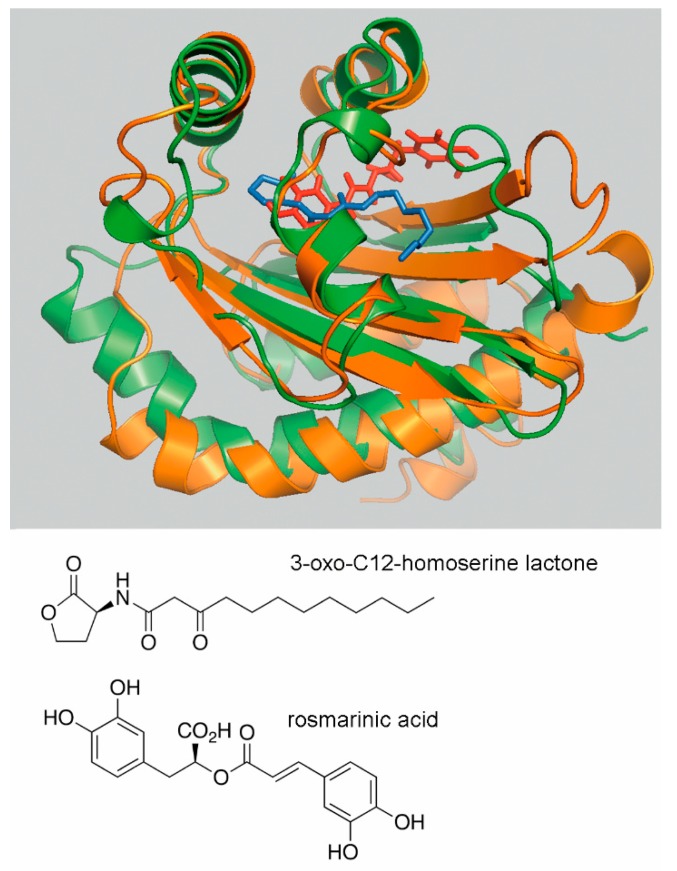
Use of virtual ligand screening to identify the plant compound rosmarinic acid as RhlR ligand. Superimposition of the ligand binding domains of LasR (green) in complex with 3-oxo-C12-homoserine lactone and the RhlR homology model (orange) onto which rosmarinic acid (RA) was docked. The LasR structure was obtained from the protein data bank (pdb ID 3IX3) and the RhlR model was generated by homology modelling as described in [83]. 3-oxo-C12-homoserine lactone and RA are shown in blue and red, respectively. The best binding position of RA with lowest glide score and glide energy is displayed. The structures of 3-oxo-C12-homoserine lactone and rosmarinic acid are shown in the lower part of the figure.

**Figure 5 ijms-19-03755-f005:**
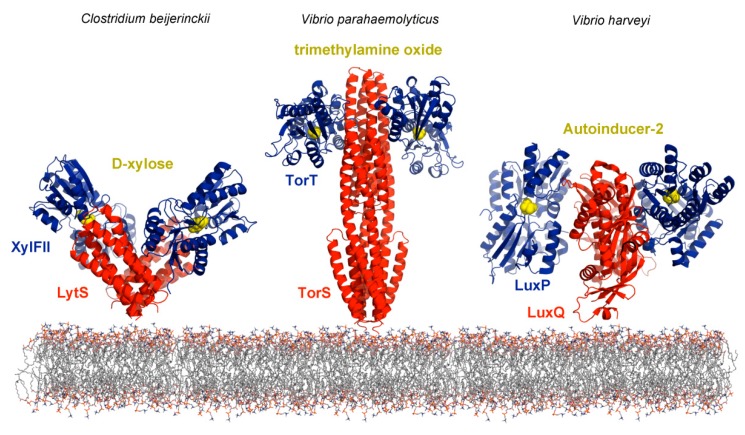
Sensor kinases that are activated by the binding of ligand loaded periplasmic binding proteins. Shown are structures of the LBDs of the sensor kinases LytS (pdb 5XSJ), TorS (pdb 3O1H) and LuxQ (pdb 2HJ9) (in red) in complex with their respective ligand-loaded binding proteins (in blue). The bacterial species and ligands bound (in yellow) are indicated.

**Table 1 ijms-19-03755-t001:** Use of high-throughput ligand screening to functionally annotate chemoreceptors. Shown are results from purified individual chemoreceptor LBDs. Data are ordered according to the LBD type. The *K*_D_ values indicated are derived mostly from microcalorimetric titrations.

Chemoreceptor Name	Bacterial Species	LBD Type	Tm without Ligand (°C)	Ligand	Tm Shift (°C)	*K*_D_ (µM)	Reference
PscD	*P. syringae*	sCACHE	56.1	acetic acid	3.3	31	[63]
				propionic acid	2.7	101	
				pyruvic acid	1.0	356	
				capric (decanoic) acid	2.2	No binding	
				glycolate	1.1	23	
McpV	*S. meliloti*	sCACHE	57.0	acetate	12.3	9.1	[64]
				propionate	12.3	3.4	
				pyruvate	11.8	33	
				glycolate	9.3	27	
				L-lactate	8.2	n.d.	
				acetoacetate	6.0	280	
				glyoxylate	5.8	n.d.	
				methyl-pyruvate	5.8	n.d.	
				α-hydroxy-butyrate	4.0	n.d.	
				α-keto-butyrate	3.8	n.d.	
PA2652	*P. aeruginosa*	sCACHE	45.5	L-malic acid	5.2	23	[65]
				citramalic acid	2.1	61	
				methylsuccinic acid	n.d.	224	
				bromosuccinic acid	3.6	1240	
				citraconic acid	2.5	210	
McpH	*P. putida*	dCACHE	47.2	adenine	3.2	2.4	[51]
				guanine	4.2	4.3	
				xanthine	3.7	2.7	
				hypoxanthine	n.d.	3.6	
				purine	n.d.	2.4	
McpU	*P. putida*	dCACHE	46.0	putrescine	11	2	[33,60]
				spermidine	2	4.5	
				cadaverine	10.5	22	
				histamine	3.7	26	
				agmatine	14	0.48	
				ethylenediamine	2.3	39	
PscA	*P. syringae*	dCACHE	40.1	L-Asp	11.0	6.1	[53]
				L-Glu	8.2	27	
				D-Asp	10.0	2.3/19 ^a^	
McpU	*S. meliloti*	dCACHE	36.0	arginine	13.1	350	[66]
				phenylalanine	12.6	53	
				proline	13.2	42 and 104	[38,66]
				tryptophan	15.0	34	
McpX	*S. meliloti*	dCACHE	45.0	trigoneline	4.8	88	[39]
				choline	9.8	0.14	
				glycine betaine	9.2	1.3	
				betonicine	1.2	2300	
				stachydrine	6.8	3.8	
				proline	4.7	45	
McpK	*P. aeruginosa*	HBM	38.7	α-ketoglutarate	5.2	301/81 ^b^	[67]
				uracil	4.2	No binding by ITC	
				γ-aminobutyrate	4.1	No binding by ITC	
				5-carbamyl phosphate	3.5	No binding by ITC	
				phenylethylamine	3.5	No binding by ITC	
				d-glucosaminic acid	3.5	No binding by ITC	
McpN	*P. aeruginosa*	PilJ	49	nitrate	3.5	47	(David Martín Mora, personal communication)

^a^: biphasic binding curve; ^b^: binding with positive cooperativity, n.d.: not determined.

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
