# Peer review of "Functional Annotation of Bacterial Signal Transduction Systems: Progress and Challenges"

_ijms, 2018, doi:10.3390/ijms19123755_

Round 1

Reviewer 1 Report

In the review entitled “Functional annotation of Bacterial signal transduction systems: Progress and challenges” by Martin-Mora et at., the authors cover with several examples the use of different experimental approaches for the characterization and annotation of regulatory proteins.

Overall, the review reads well and the chapters contain the proper literature. The authors describe examples of this functional annotation using four main approaches. However, I have a few suggestions to improve the quality of the review:

1-     A whole new section should be added covering the vast knowledge of crystal structures solved in the presence of ligands and how this information can be used to predict the function of an unknown regulatory protein. This is crucial, since the linear sequence may not be the best way to compare two regulatory proteins but the overall 3D structure. There are numerous cases where the identity of regulatory protein share low identity but their folding (3D structures, models, etc) are much more similar. The authors should explore this avenue, since many regulators share common ligand pockets and the residues involved in this interaction may or may not be conserved.

2-     Similar to the previous comment, a whole section involving regulatory processes through protein-protein interaction is necessary. For example, in section 5.2 it would be good if the authors add a comment about the different methods of studying protein-protein interaction and the role of regulatory ligands.

3-     In the section 1 “Functional annotation using in vivo approaches”, the review seems to be too negative with this approach and mostly described unsuccessful cases. I would add more examples where this approach can be successfully used. In addition, I would add a paragraph stating the cases where the approach did not solve the function of chemoreceptor.

4-     The authors mention the study of the protein thermal stability to analyze protein-ligand interactions. However, they only cover the cases where the ligand increases the melting temperature of the protein. There are cases where, the binding of the ligand decreases the thermal stability of the protein and some interpretation has given to this phenomena. I would recommend to see this article https://www.sciencedirect.com/science/article/pii/S0006349508784649?via%3Dihub

Minor revisions:

1-     There are numerous cases where the names of the microorganisms and genes are not in italics. Please correct this issue.

2-     Figure Legend 1: LBP description is located in between a comma and a semi colon, please be consistent through all the text.

3-     Line 502. Title for section 5.1 is not necessarily related to the content of the section. Please modify.

4-     Line 303, replace “generated” by isolated or cloned.

5-     Line 361. Double space before pstS.

Author Response

Detailed response to the reviewers’ comments

Reviewer 1:

In the review entitled “Functional annotation of Bacterial signal transduction systems: Progress and challenges” by Martin-Mora et at., the authors cover with several examples the use of different experimental approaches for the characterization and annotation of regulatory proteins.

Overall, the review reads well and the chapters contain the proper literature. The authors describe examples of this functional annotation using four main approaches. However, I have a few suggestions to improve the quality of the review:

A whole new section should be added covering the vast knowledge of crystal structures solved in the presence of ligands and how this information can be used to predict the function of an unknown regulatory protein. This is crucial, since the linear sequence may not be the best way to compare two regulatory proteins but the overall 3D structure. There are numerous cases where the identity of regulatory protein share low identity but their folding (3D structures, models, etc) are much more similar. The authors should explore this avenue, since many regulators share common ligand pockets and the residues involved in this interaction may or may not be conserved.

Response: This suggestion has been taken up. The following section was added to the manuscript:

“4. Three-dimensional structural information provides clues on ligands recognized by homologous sensor proteins

Ligand binding domains are characterized by a high degree of sequence divergence and frequently the overall sequence similarity between homologous proteins is not reflected in a similarity of ligands recognized [23]. However, three-dimensional structural information is now an invaluable tool to get initial information on ligands recognized. We would like to illustrate this issue on the example of CACHE domains that exist either in a mono-modular (sCACHE) or bi-modular (dCACHE) configuration [68]. CACHE domains are the most abundant sensor domains in both, histidine kinases [69] and chemoreceptors [7].

As significant number of structures deposited in the Protein Data Bank (pdb) are the result of structural genomics initiatives and represent a source of information on the function of homologous proteins. In the framework of structural genomic projects on Anaeromyxobacter dehalogenans and Vibrio parahaemolyticus, structures of sCACHE domains have been deposited at pdb (pdb ID 4K08 and 4EXO, respectively). In both cases, their binding pocket was occupied by a ligand, namely acetate in 4K08 and pyruvate in 4EXO. In the case of 4K08 structural model, acetate was a component of the crystallization buffer. Since pyruvate was absent from the crystallization buffer of the 4EXO structure, the ligand must have been co-purified with the protein. P. putida KT2440 has a single chemoreceptor with a sCACHE domain (PP_2861), which shares only 22 % sequence identity with the 4EXO protein. The above observations have directed our research and we have tested whether the LBD of PP_2861 also bound acetate and pyruvate. We were able to show that this was the case and have identified with propionate and L-lactate two additional ligands. The corresponding chemoreceptor was renamed McpP and was found to mediate chemoattraction to these four ligands [61].

Similarly, structural information was also useful to study dCACHE domain. The first structure of a dCACHE domain deposited into pdb was entry with ID 3C8C; solved in a structural genomics project of Vibrio cholerae. In analogy to the case above, the binding pocket of this structure was occupied with a ligand, L-Ala, and since L-Ala was not present in the crystallization buffer, it must have been co-purified with the protein suggesting that it is a physiological ligand. This information has been useful to identify other amino acid sensing chemoreceptors and it was established that dCACHE containing chemoreceptors form the primary family of amino acid responsive chemoreceptors [18].

There are different sub-families within dCACHE domains, namely those that bind C4-dicraboxylic acids like succinate and malate [70, 71] or those that bind different amines such as amino acids, GABA, polyamines, taurine, purines or quaternary [23, 33, 36, 39, 51, 60, 72-75]. Structural information has now provided the molecular detail of C4-dicarboxylic acid [70, 71] and amine recognition at different dCACHE domains [33, 73-75] (Fig. 3). In all cases the ligand is bound to the membrane distal module. The comparison of amine recognition patterns in structures from different species (P. aeruginosa, P. putida, Vibrio cholerae, C. jejuni), and in complex with different ligands (putrescine, histamine, taurine, amino acids) identifies the consensus motif Y-x-D-x(n)-D that coordinates the amino group of ligands through a hydrogen bonding network. Therefore, the sequence analysis of dCACHE domains of unknown function for the presence of this motif can provide first clues on the ligand recognized.”

In addition, a new Figure (Fig. 3) was prepared for illustration.

Similar to the previous comment, a whole section involving regulatory processes through protein-protein interaction is necessary. For example, in section 5.2 it would be good if the authors add a comment about the different methods of studying protein-protein interaction and the role of regulatory ligands.

Response: We see the point of this referee but we do not want to go too much into mechanistic issues on the action ligand binding proteins that would, in our opinion, distract from the main lines of this review. The messages that we wanted to bring across in this section are: (1) Binding protein mediated stimulation occurs in sensor kinases and chemoreceptors. (2) Ligand binding proteins interact with different types of sensor domain and are not restricted a LBD type, and (3) Such mechanisms appear to be more frequent than generally assumed and the few examples available are rather due to difficulties in identifying these proteins.

We have taken up the suggestion of this referee and have added the following sentence on the different approached pursued to study protein-protein interaction. “Approaches for large scale characterization of protein-protein interactions include yeast two-hybrid screening [106], co-immunoprecipitation [107], affinity purification-mass spectrometry [108], in vitro cross linking-mass spectrometry [109], phage display [110], yeast display [111] or reprogramming yeast mating [112].”

In the section 1 “Functional annotation using in vivo approaches”, the review seems to be too negative with this approach and mostly described unsuccessful cases. I would add more examples where this approach can be successfully used. In addition, I would add a paragraph stating the cases where the approach did not solve the function of chemoreceptor.

Response: The referee is absolutely right there. This negativity was not intended. We now state that genomic approaches have led to the identification of many chemoreceptors and cite representative articles. The reasons why such approaches may not be successful are discussed below and consist in the possibility that there are multiple receptors, the inducible nature of chemotaxis or masking by energy taxis. The following sentences have been added “Using this approach the function of many chemoreceptors has been identified. As representative examples we would like to cite here the identification of chemoreceptors for naphthalene [24], cyclic carboxylic acids [25], cytosine [26], inorganic phosphate [27] or boric acid [28].”

The authors mention the study of the protein thermal stability to analyze protein-ligand interactions. However, they only cover the cases where the ligand increases the melting temperature of the protein. There are cases where, the binding of the ligand decreases the thermal stability of the protein and some interpretation has given to this phenomena. I would recommend to see this article https://www.sciencedirect.com/science/article/pii/S0006349508784649?via%3Dihub

Response: We agree. The possibility that ligand destabilize proteins, although much less frequent, is now mentioned. Takin into account comments by referee 2 we have rearranged this section and have transferred information on the assay, previously in the action of the AdmX regulator, to the beginning of the section 3. We now also mention that in some cases ligand binding destabilizes the protein and introduce the concept of ligand binding to the native or a unfolded form of the protein. The article mentioned by this referee is cited. This section has been changed into “This method is based on the fact that the binding of ligands to proteins either increases or decreases their thermal stability. Typically Differential Scanning Fluorimetry [49] is used for Thermal Shift Assays. In these assays, a hydrophobic fluorescent dye is added to the protein that is then exposed to a temperature gradient. In soluble proteins, hydrophobic amino acids are typically in the protein interior and excluded from the solvent. However, during temperature-induced protein unfolding, these amino acids are exposed to the solvent leading to additional fluorescent dye binding altering in turn protein fluorescence. From the recorded fluorescence changes, the melting temperature (Tm value) can be derived, which corresponds to the temperature at which half the protein is unfolded, whereas the remaining half is still in its folded state. Typically, the binding of a ligand to a protein retards protein unfolding and as such increases the Tm value. However, in some cases a reduction in Tm is observed indicating that the ligand binds to an unfolded state of the protein [50, 51]. There are cases where the binding of different ligands to the same protein causes either increases or decreases of the thermal stability [50].”

Minor revisions:

There are numerous cases where the names of the microorganisms and genes are not in italics. Please correct this issue.

Response: This has also been mentioned by referee 2. However, all gene and bacterial species names were in the italics in the word document that I have uploaded. The manuscript has then been reformatted by the editorial office and as a consequence in numerous places gene and species names appeared in regulator font in the manuscript that was sent out to the referees. I have carefully revised the manuscript to change back the format into italics at the corresponding places.

Figure Legend 1: LBP description is located in between a comma and a semi colon, please be consistent through all the text.

Response: Thanks. Done.

Line 502. Title for section 5.1 is not necessarily related to the content of the section. Please modify.

Response: We agree. This heading has been changed to “Sensor Protein Activation By The Binding Of Ligand Binding Proteins”

Line 303, replace “generated” by isolated or cloned.

Response: OK. “generated” has been replaced by “isolated”

Line 361. Double space before pstS.

Response: Thanks. Done.

Reviewer 2 Report

Sensing proteins play essential roles in bacteria-host, bacteria-environment interactions and regulation of bacterial physiology. Progress has been made on identifying those sensing proteins based on genetic mutants, bioinformatics analyses, etc. On the other hand, the lack of knowledge on signals recognized by sensor proteins represents a major bottleneck in the signal transduction research field, as the authors have pointed out in this nicely written review. 

Experimental approaches seem to be a logic solution and primary strategy to identify the signals for the sensor proteins. This review by Martin Mora et al. summarizes over the last 10 years, the experimental approaches and techniques applied in the filed of signal identification of the sensor proteins, and some of the progresses made in the field. This review is very much needed the signal transduction field, which is apparently trying to catch up with high-throughput studies frequently developed in many other research fields. 

I have a few comments about the overall organization of this review.

line 80, “2. Functional Annotation Using in Vivo Approaches”. 

Functional studies on ligand characterization using genetic approaches have been quite successful in the past and examples of successful researches are substantial. In the current version, this is very briefly covered only in lines 81-83, which is also used for the transition. I suggest that before getting into potential challenges of the genetic approaches (which could possibly be moved to section 5), the authors could expand the discussion about past successful studies using genetic approaches since the title emphasizes both “progress and challenges”.  

Line 123,  “3. Functional Annotation Using Purified Recombinant Sensor Proteins”

For this section, the overall layout is bit unclear, section 3.1 is a specific example of AdmX and the ligand IAA identification using ITC; section 3.2 is discussion about thermal shift assays, its potential challenges.  

Here are my thoughts, if this section is about techniques facilitating in vitro studies using recombinant proteins, I will suggest moving thermal shift assays from 3.2 to 3.1; the new 3.2 could be discussions about ITC and other techniques; and then 3.3. could be the example of AdmX and IAA identification.

line 242, “4. Protein Structure Based Virtual Ligand Screening”

In this section, the authors used a plant-derived compound, Rosmarinic acid” as an example for the virtual ligand screening method. This example itself is very interesting and a perfect fit, and is worth being highlighted. However, since this is a broad topic review, it will be good to discuss more on the rational and the methodology of virtual ligand screening, and include other successful examples as well, in addition to RA discovery.

minor comments:

line 80, 2. “Functional Annotation Using in Vivo Approaches”  using phenotypic analyses of genetic mutants is more precisely genetic approaches, it is a subtle difference, I will suggest using genetic approaches in the subtitle.

line 99, P. aeruginosa is not italicized, same in a few other places, line 96, 104, etc. 

line 303, LBDs, mostly from different Pseudomonas species.

Author Response

Reviewer 2

Sensing proteins play essential roles in bacteria-host, bacteria-environment interactions and regulation of bacterial physiology. Progress has been made on identifying those sensing proteins based on genetic mutants, bioinformatics analyses, etc. On the other hand, the lack of knowledge on signals recognized by sensor proteins represents a major bottleneck in the signal transduction research field, as the authors have pointed out in this nicely written review. 

Experimental approaches seem to be a logic solution and primary strategy to identify the signals for the sensor proteins. This review by Martin Mora et al. summarizes over the last 10 years, the experimental approaches and techniques applied in the filed of signal identification of the sensor proteins, and some of the progresses made in the field. This review is very much needed the signal transduction field, which is apparently trying to catch up with high-throughput studies frequently developed in many other research fields. 

I have a few comments about the overall organization of this review.

line 80, “2. Functional Annotation Using in Vivo Approaches”. 

Functional studies on ligand characterization using genetic approaches have been quite successful in the past and examples of successful researches are substantial. In the current version, this is very briefly covered only in lines 81-83, which is also used for the transition. I suggest that before getting into potential challenges of the genetic approaches (which could possibly be moved to section 5), the authors could expand the discussion about past successful studies using genetic approaches since the title emphasizes both “progress and challenges”.  

Response: This has also been mentioned by referee 1. We now state that this approach has resulted in the functional annotation of a large number of chemoreceptors and cite 5 representative examples. The following sentences have been added “Using this approach the function of many chemoreceptors has been identified. As representative examples we would like to cite here the identification of chemoreceptors for naphthalene [24], cyclic carboxylic acids [25], cytosine [26], inorganic phosphate [27] or boric acid [28].”

Line 123,  “3. Functional Annotation Using Purified Recombinant Sensor Proteins”

For this section, the overall layout is bit unclear, section 3.1 is a specific example of AdmX and the ligand IAA identification using ITC; section 3.2 is discussion about thermal shift assays, its potential challenges.  

Here are my thoughts, if this section is about techniques facilitating in vitro studies using recombinant proteins, I will suggest moving thermal shift assays from 3.2 to 3.1; the new 3.2 could be discussions about ITC and other techniques; and then 3.3. could be the example of AdmX and IAA identification.

Response: This section has been rearranged. As suggested, information on the thermal shift assays were moved up to explain the technique. This is then illustrated by the AdmX transcriptional regulator followed by work on chemoreceptors. We wish to maintain this order since the experimental data shown on AdmX (Fig. 2) illustrate well the approach.

line 242, “4. Protein Structure Based Virtual Ligand Screening”

In this section, the authors used a plant-derived compound, Rosmarinic acid” as an example for the virtual ligand screening method. This example itself is very interesting and a perfect fit, and is worth being highlighted. However, since this is a broad topic review, it will be good to discuss more on the rational and the methodology of virtual ligand screening, and include other successful examples as well, in addition to RA discovery.

Response: We agree. We have added the following section to illustrate the usefulness of virtual ligand screening in the identification of ligands for bacterial sensor proteins:

 “5.2. Further examples that illustrate the usefulness of virtual ligand screening to identify protein ligands

A number of in silico docking experiments have been reported that have led to the identification of ligands for a number of bacterial signal transduction systems. Two studies report the identification of ligands that bind to the transcriptional regulators AphB of Vibrio cholerae [88, 89] that regulates the expression of genes encoding cholera toxin and toxin-co-regulated pilus. In both cases virtual docking results were verified experimentally by NMR [88] and/or ITC [89]. One of the compounds identified, ribavirin, was shown to inhibit cholera toxin production, which in turn was reflected in a reduction of intestinal colonization and intracellular survival of V. cholerae [89].

The sensor kinase KinB controls the synthesis of the exopolysaccharide alginate in P. aeruginosa. It has a periplasmic LBD that forms a four-helix bundle structure, but the nature of ligands that bind to KinB-LBD is unknown. In silico ligand screening assays were conducted that resulted in the identification of several sugar phosphates, including compounds that serve as precursors and intermediates of the alginate biosynthesis. The binding of two of these compounds was verified experimentally using biolayer interferometry [90].

AldR is a transcriptional regulator of Mycobacterium tuberculosis and controls the expression of the ald genes encoding the alanine dehydrogenase that plays a major role in the response to nutrient starvation. AldR is composed of a DNA binding domain and an AsnC type LBD [91]. To identify AldR ligands, in silico screening assays identified a tetrahydroxyquinolone carbonitrile derivative. This compound was found to bind to purified AldR and to prevent protein binding to promoter DNA [91].

LdtR is a transcriptional regulator from Liberibacter asiaticus, a non-culturable citrus fruit pathogen. Previous work using thermal shift assays has resulted in the identification of several LtdR ligands that were found to reduce LdtR-DNA binding [92]. To study of the effect of the compounds identified, the authors used S. meliloti and Liberibacter crescens that possess LdtR homologues and showed that molecules identified decrease their stress tolerance. However, no information was available on the binding site of these ligands at LdtR. In silico docking studies with one of the molecules identified, benzbromarone, identified its binding pocket at LdtR for [93]; a finding that was verified by site-direct mutagenesis. The gained knowledge will be useful for the design of therapeutics to fight this pathogen.”

minor comments:

 line 80, 2. “Functional Annotation Using in Vivo Approaches”  using phenotypic analyses of genetic mutants is more precisely genetic approaches, it is a subtle difference, I will suggest using genetic approaches in the subtitle.

Response: We agree. This change has been made.

line 99, P. aeruginosa is not italicized, same in a few other places, line 96, 104, etc. 

Response: In the initial word file I have uploaded, all names of genes and bacterial species were in italics. The manuscript has been reformatted by the editorial team of the journal and as a consequence gene and bacterial species do not appear any more in italics in many places. I have revised the entire manuscript to change the format back to italics.

line 303, LBDs, mostly from different Pseudomonas species.

Response: Thanks. Done.

Round 2

Reviewer 1 Report

Thanks for the modifications. I do have a few minor changes:

Minor Comments

Line 132. Replace "sift" for "shift"

Line 312. Delete "model"

Line 314. delete "." after protein

Line 329. Replace "dicraboxylic" for "dicarboxylic"

Author Response

Dear editor,

Many thanks for your email. Please find the revised version of this manuscript.

These are the change made in response to this referee.

Minor Comments

Line 132. Replace "sift" for "shift"

Response: Thanks. Done.

Line 312. Delete "model"

Response: “In the case of 4K08 structural model...” was replaced by “In the case of the 4K08 structure…”

Line 314. delete "." after protein

Response: The “.” marks the end of a sentence and was therefore left as it is. The sentence reads “Since pyruvate was absent from the crystallization buffer of the 4EXO structure, the ligand must have been co-purified with the protein.”

Line 329. Replace "dicraboxylic" for "dicarboxylic"

Response: Thanks. Done.

Please do not hesitate in contacting if there are any further issues with this manuscript.

I look forward to hearing from you.

Yours sincerely,

Tino Krell
